Expression of the IL-18-related gene PTX3 correlates with clinicopathological features and prognosis in glioma patients

Wang Delin 1
Liu Cuimei 2
Sun Bohao 3
Zhang Xiaodong 1
Zhou Yejun 1
Hu Zhonglin 1
Cao Duanzheng 1
Zhang Jing 2522017@zju.edu.cn 3
Xu Jinfang xujinfang@zju.edu.cn 4
1 Department of Neurosurgery, Jiande First People’s Hospital , Hangzhou , Zhejiang , China
2 Department of Neurology, Jiande First People’s Hospital , Hangzhou , Zhejiang , China
3 Department of Pathology, Second Affiliated Hospital, School of Medicine, Zhejiang University , Hangzhou , Zhejiang , China
4 Department of Neurosurgery, The Second Affiliated Hospital Zhejiang University School of Medicine , Hangzhou , Zhejiang , China
Haraguchi Tokuko
Electronic publication date: 2025 Jul 10
Publication date: 2025
Volume: 13
Electronic Location ID: e19675
Received 2025 Feb 19; Accepted 2025 Jun 9
Copyright: ©2025 Wang et al.
Copyright year: 2025
Copyright holder: Wang et al.
License: This is an open access article distributed under the terms of the Creative Commons Attribution License, which permits unrestricted use, distribution, reproduction and adaptation in any medium and for any purpose provided that it is properly attributed. For attribution, the original author(s), title, publication source (PeerJ) and either DOI or URL of the article must be cited.
License URL: https://creativecommons.org/licenses/by/4.0/

Keywords: PTX3, Glioma, Poor-prognosis predictor, Clinical stage, Immune cell infiltration

Funding: The Science and Technology Development Plan of Jiande 2023SJZX12 The Zhejiang Provincial Medical and Health Science and Technology Project 2024KY292 This work was funded by grants from the Science and Technology Development Plan of Jiande (grant no. 2023SJZX12) and the Zhejiang Provincial Medical and Health Science and Technology Project (grant no. 2024KY292). The funders had no role in study design, data collection and analysis, decision to publish, or preparation of the manuscript.

==============================
Background

Glioma, a highly aggressive brain tumor, presents significant challenges in prognosis and treatment. This study investigates the role of PTX3 expression in glioma and its correlation with patient outcomes, addressing a gap in current research regarding its molecular mechanisms.

Materials and Methods

RNA sequencing data and clinical information for glioma patients were obtained from The Cancer Genome Atlas (TCGA). A multigene prognostic signature based on IL-18 signaling-related genes (ISRGs) was constructed using the least absolute shrinkage and selection operator (LASSO) Cox regression method. The functional roles of PTX3 were analyzed through Gene Ontology (GO), Kyoto Encyclopedia of Genes and Genomes (KEGG), and Gene Set Enrichment Analysis (GSEA). Single-sample GSEA (ssGSEA) was used to assess the association between PTX3 expression and immune cell infiltration. The relationship between PTX3 expression and clinicopathological features was also examined. Prognostic relevance was evaluated using univariate and multivariate Cox regression models, and Kaplan–Meier survival analysis was performed. PTX3 protein expression was validated via immunohistochemistry in 56 glioma specimens.

Results

The LASSO Cox regression model identified a nine-gene prognostic signature, including BMP2, NCF1, HSPB1, PIGT, PTX3, CCNA2, CCNB2, CCN4, and DES. Functional enrichment analysis revealed that PTX3-associated differentially expressed genes were significantly enriched in pathways such as cytokine–cytokine receptor interaction and PI3K-Akt signaling, which are critical for immune response and cell proliferation in glioma. PTX3 expression showed a strong correlation with immune cell infiltration, particularly macrophages, neutrophils, T cells, and natural killer cells, suggesting a role in modulating the tumor microenvironment. Pan-cancer analysis indicated that PTX3 is markedly upregulated in various cancers, especially gliomas, highlighting its potential as a biomarker. PTX3 expression was also associated with clinical features such as WHO grade, IDH mutation status, and 1p/19q co-deletion, with higher PTX3 levels linked to poorer survival outcomes. Immunohistochemistry confirmed elevated PTX3 protein expression in both lower-grade glioma and glioblastoma multiforme.

Conclusions

These findings highlight the critical role of PTX3 in glioma and suggest its potential as both a prognostic biomarker and therapeutic target. This study provides a foundation for future research into targeted therapies involving PTX3.

Introduction

Gliomas, characterized as highly aggressive brain tumors, present substantial challenges in terms of patient prognosis and treatment options, placing considerable strain on both individuals and healthcare systems (Galldiks, Langen & Pope, 2015; Thenuwara, Curtin & Tian, 2023; Zhao et al., 2024). The invasive nature of gliomas, along with their poor clinical outcomes, underscores the urgent need for a deeper understanding of their molecular mechanisms, which is crucial for developing innovative therapeutic approaches (Giese et al., 2003; Salhia et al., 2006; Wang et al., 2022a; Wang et al., 2022b). Previous studies have identified several molecular markers associated with glioma (Michaëlsson et al., 2023; Žugec et al., 2024); however, the specific roles and mechanisms of Pentraxin 3 (PTX3) remain inadequately explored, highlighting a significant gap in the current research. This study aims to address this gap by evaluating the expression levels of PTX3 and its correlation with patient prognosis in glioma. The findings may provide insights into the identification of novel prognostic biomarkers and therapeutic targets. Understanding the role of PTX3 in glioma progression is crucial, as it could inform the development of personalized treatment strategies and improve patient outcomes in this complex disease context.

The role of IL-18 in glioma is complex and dynamic, significantly influencing tumor biology through multiple mechanisms. On one hand, IL-18 enhances the activity of cytotoxic T lymphocytes and natural killer (NK) cells, thereby boosting immune responses against tumor cells (Ohno et al., 2025; Shen et al., 2025). Research has established a strong association between IL-18 expression and glioma progression, particularly within the tumor microenvironment, where it stimulates macrophages and other immune cells to release pro-inflammatory cytokines, thereby promoting tumor cell proliferation and metastasis (Ji et al., 2024; Sun et al., 2023; Thanasupawat et al., 2024). Moreover, evidence suggests that IL-18 may influence glioma growth by modulating the inflammatory microenvironment associated with tumors. Specifically, microglia in the glioma microenvironment become activated and produce IL-18, which in turn enhances the migratory capacity of glioma cells (da Silva et al., 2020; Thanasupawat et al., 2024; Yeh et al., 2012). In terms of therapeutic strategies, IL-18 antagonists have demonstrated the potential to inhibit tumor growth in experimental models, indicating that modulating IL-18 levels or its signaling pathways may offer novel approaches to enhance treatment outcomes for glioma patients (da Silva et al., 2020; Ji et al., 2024). However, the precise mechanisms by which IL-18 functions in glioma require further investigation to elucidate its dual roles in tumor biology and immune responses, thereby informing clinical treatment strategies.

This study investigates the expression levels of PTX3, a key component of the IL-18 signaling pathway, and its implications for the prognosis of glioma patients. PTX3 is an acute-phase protein secreted by immune cells that plays a critical role in immune responses, inflammation, and tissue repair (Bottazzi et al., 2016; Kofla-Dlubacz et al., 2024; Xu et al., 2025). PTX3 expression varies across different cancers; for example, it is elevated in certain tumors such as hepatocellular carcinoma (Cabiati et al., 2022; Chen et al., 2024; Deng et al., 2020), ovarian cancer (Liu et al., 2024; Naylor et al., 2024; Tanılır Çağıran et al., 2024), and bladder cancer (Goodison et al., 2012; Vikerfors et al., 2024). This differential expression highlights the potential of PTX3 as a therapeutic target and provides insights into the biological mechanisms within the tumor microenvironment. However, the relationship between PTX3 expression, clinical factors, and prognosis in glioma remains unclear, warranting further investigation to clarify its role in cancer progression. Our comprehensive study identifies potential genetic biomarkers, offering novel insights into therapeutic targets for glioma.

Although previous studies have identified several molecular markers relevant to glioma, the precise functions and mechanisms of PTX3 remain insufficiently understood, highlighting a significant gap in the existing literature. Our study aims to address this gap by exploring the potential of PTX3 as both a prognostic biomarker and a therapeutic target in glioma. We focus on the expression profiles of IL-18 signaling-related genes (ISRGs) and their functional implications in glioma. Previous research has demonstrated the critical role of IL-18 in the tumor immune microenvironment, suggesting that PTX3 may be crucial in modulating the immune response in glioma (Cheng et al., 2024; Saetang et al., 2020). The relationship between PTX3 expression and various clinical parameters further underscores its potential as a novel prognostic marker, which could improve patient stratification and inform treatment strategies.

This study utilizes a multifaceted approach, including RNA sequencing analysis, functional enrichment analysis, least absolute shrinkage and selection operator (LASSO) regression modeling, and immunohistochemical evaluation, to investigate the relationship between PTX3 expression and glioma prognosis, as well as its impact on immune cell infiltration. In conclusion, this research aims to address the existing gap in PTX3 studies related to glioma by employing diverse bioinformatics techniques to explore its role in glioma progression, thus providing new insights and evidence for potential therapeutic strategies.

Materials and Methods

Gene expression data processing and normalization

RNA sequencing data for glioblastoma (GBM), lower-grade glioma (LGG), and normal brain tissues were obtained from The Cancer Genome Atlas (TCGA) and GTEx through UCSC XENA (https://xenabrowser.net/datapages/). The dataset included 174 GBM cases and 532 LGG samples. Genes with expression levels below 1 in more than 50% of the samples were excluded from the analysis. The validation cohorts, which included comprehensive expression profile data from GSE50161 and GSE7696, were sourced from the GEO database (https://www.ncbi.nlm.nih.gov/gds). The prognostic validation cohort is derived from the Chinese Glioma Genome Atlas (CGGA) database (http://www.cgga.org.cn). Additionally, 256 genes associated with the IL-18 signaling pathway, collectively termed ISRGs, were curated from the WP_IL18_SIGNALING_PATHWAY database, as outlined in Table S1.

Development of protein-protein interactions and identification of central genes

The protein-protein interaction (PPI) network of differentially expressed genes (DEGs) was constructed using STRING (http://string-db.org). The network was then visualized using Cytoscape software (https://www.cytoscape.org).

Functional enrichment analysis

Gene Ontology (GO) and Kyoto Encyclopedia of Genes and Genomes (KEGG) enrichment analyses were conducted using the clusterProfiler package from Bioconductor (Yu et al., 2012). Gene expression data normalization and identification of differentially expressed genes (DEGs) were performed using the limma package in R (version 3.18) (Li et al., 2021; Wang et al., 2019). The criteria for functional enrichment analysis of DEGs were set as —logFC— > 2 and an adjusted P-value < 0.05.

Gene set enrichment analysis

The clusterProfiler R package (version 3.14.3) was used to examine functional and pathway differences between groups with varying PTX3 expression levels. Significantly enriched hallmarks were identified based on an FDR q-value < 0.05.

Machine learning algorithm to screen key genes

To identify key biomarkers associated with glioma prognosis, we extracted data from the TCGA database and developed gene expression profiles. Using the LASSO algorithm from the glmnet R package, we conducted a comprehensive screening for significant biomarkers. A total of 256 ISRGs were identified through LASSO Cox regression analysis, aimed at minimizing redundancy and preventing model overfitting. The optimal penalty parameter, lambda, was determined through a 10-fold cross-validation procedure. Nine genes were ultimately selected to construct a prognostic risk score model for predicting overall survival (OS) in glioma patients.

Correlation analysis between PTX3 expression levels and immune cell infiltration

The single-sample Gene Set Enrichment Analysis (ssGSEA) algorithm, implemented in the “GSVA” (version 1.34.0) R package, was used to assess the infiltration levels of 28 distinct immune cell types in the tumors. Subsequently, Spearman’s correlation analysis was performed to explore the associations between PTX3 expression levels and immune cell infiltration status.

Correlation analysis between PTX3 expression levels and clinicopathological characteristics

Clinicopathological data for patients diagnosed with glioblastoma multiforme GBM and LGG, including OS, disease-specific survival (DSS), and progression-free interval (PFI), were obtained from the TCGA-GBM and TCGA-LGG projects. Nomograms that integrate clinical features alongside PTX3 models were constructed utilizing the “rms” R package (version 6.3) to forecast the overall survival (OS) of glioma samples. Logistic regression analysis was performed to assess the relationship between PTX3 expression levels and the clinicopathological characteristics of glioma patients.

Construction of transcription factor-gene interaction networks

The hub genes identified previously were subsequently uploaded to NetworkAnalyst (version 3.0, https://www.networkanalyst.ca/) to develop transcription factor (TF) and gene interaction networks. The construction of the TF-gene interaction network was facilitated through the utilization of the JASPAR database (http://jaspar.genereg.net) via the NetworkAnalyst platform.

Construction of the TF-miRNA co-regulatory network

Data regarding the TF-miRNA coregulatory network was sourced from the RegNetwork database (http://www.regnetworkweb.org/), which amalgamates established regulatory interactions from various databases along with prospective regulations inferred from TF binding sites. The visualization of the TF-miRNA coregulatory network was accomplished utilizing the NetworkAnalyst platform.

Ethics statement

Glioma tissues and adjacent paracancerous tissues were obtained from the Department of Pathology, Second Affiliated Hospital, School of Medicine, Zhejiang University. This study adhered to the ethical principles outlined in the Declaration of Helsinki. Given its retrospective nature, patient consent was waived for retrospective data by the Ethics Committee. We randomly selected glioma and adjacent normal brain tissues from 108 patients who had undergone surgical resection at the Second Affiliated Hospital, Zhejiang University. The research protocols received endorsement from the Ethics Committee of the Second Affiliated Hospital, Zhejiang University School of Medicine, located in Hangzhou, China (approval number: 2024-0649).

Immunohistochemical staining

Antigen retrieval was performed using an autoclave after tissue sections were dewaxed in water. The sections were equilibrated to room temperature and then treated with 3% hydrogen peroxide for 10 min. To block non-specific binding, a 10% serum solution was applied. The sections were incubated overnight at 4 °C with the primary antibody, PTX3 (Proteintech, dilution 1:1000). The following day, the sections were incubated with the secondary antibody at room temperature for one hour. Positive signals were visualized in brown using DAB as the chromogenic substrate. The nuclei were counterstained with hematoxylin for 30 s. Finally, the sections were mounted with a neutral dendrimer, examined microscopically, and photographed for documentation. The staining outcomes were systematically evaluated by a duo of pathologists, who assigned scores according to specified criteria: tumor intensity was categorized as 0 (negative), 1 (weak positive), 2 (moderate positive), or 3 (strong positive); tumor extent was classified as 0 (0–10%), 1 (10%–25%), 2 (26%–50%), 3 (51%–75%), or 4 (76%–100%). The overall score was determined by multiplying the intensity score with the extent score. Scores between six and 12 were interpreted as indicative of elevated PTX3 expression or positive status, while scores from 0 to five denoted low expression or the absence of PTX3, thus indicating a negative outcome.

Cell culture and transfection procedures

The U251 cell line was acquired from the American Type Culture Collection (ATCC) (Manassas, VA, USA). U251 cells were maintained in DMEM/F12 medium (Gibco, Waltham, MA, USA), which was enriched with 10% fetal bovine serum (Lonsera, Australia) and 1% penicillin/streptomycin (Beyotime Biotechnology, Jiangsu, China). Following this, the silencing of PTX3 was achieved through cell transfection. Small interfering RNA (siRNA) that specifically targets PTX3 (designated as si-PTX3) was procured alongside a negative control (si-NC) from He Yuan Biology (Shanghai, China). The sequence for si-PTX3 was identified as GGGAUAGUGUUCUUAGCAATT. The transfection of U251 cells was performed utilizing Lipofectamine 3000 (Invitrogen, Thermo Fisher Scientific, Waltham, MA, USA) according to the manufacturer’s protocols, with a transfection duration of 48 h.

Quantitative real-time PCR

Total RNA extraction was conducted employing the TRIzol reagent produced by Invitrogen (Waltham, MA, USA). The complementary DNA (cDNA) synthesis was carried out utilizing the PrimeScript RT kit provided by Takara (Shiga, Japan). The mRNA expression levels were quantified through the application of qRT-PCR. The primers were meticulously designed by Qingke (Hangzhou, China), specifically: PTX3 forward primer: 5′-GCATAATAGGAACACTTGAGAC-3′ and reverse primer: 5′-CTGACAGAGACACAGCATT-3′.

Wound healing assay

In the wound healing assay, cells were cultivated in 24-well plates. A sterile tip was utilized to create scratches in the cells, aligned perpendicularly to the previously marked line. Following the scratching process, the resulting wound area was captured using a light microscope, and cell migration was assessed at designated time intervals of 0 and 24 h. Each experiment was conducted in triplicate to ensure reliability of the results.

Cell proliferation assay

Cells were harvested via centrifugation and subsequently resuspended in the culture medium to achieve a concentration of 104 cells per milliliter. Following this, 200 µL of the culture medium was dispensed into each well of a 96-well plate. The evaluation of cell proliferation was performed utilizing the CCK-8 assay (Beyotime Biotechnology, Shanghai, China), with measurements taken at 0, 24, 48, and 72 h in accordance with the manufacturer’s guidelines. The optical density at 450 nm (OD450) was quantified and documented using GraphPad Prism software to determine the proliferative capacity of the cells. Each experimental procedure was conducted in triplicate.

Colony formation assay

In order to conduct the colony formation assay, transfected U251 cells were seeded into a 6-well plate at a density of 500 cells per well and cultured for a duration of 14 days. Following this incubation period, the resulting colonies were fixed using a 4% paraformaldehyde solution (Beyotime, Shanghai, China) for 30 min and subsequently stained with a 0.1% crystal violet solution (C0121; Beyotime, Shanghai, China). Ultimately, the colonies were counted to facilitate further analysis.

Statistical analysis

Bioinformatics analyses were conducted using R software (version 4.2.0). A one-way ANOVA was performed to assess differences among multiple groups. The Wilcoxon rank-sum test was conducted for comparing two groups.

Results

The mRNA expression profiles of selected ISRGs in glioma

To identify the most suitable candidate genes, we employed a LASSO Cox regression method to develop a gene signature based on ISRGs. This analysis resulted in the creation of a prognostic model consisting of nine genes with non-zero coefficients: BMP2, NCF1, HSPB1, PIGT, PTX3, CCNA2, CCNB2, CCN4, and DES (Fig. 1A). A univariate Cox regression analysis was conducted to assess the correlation between ISRG expression levels and OS in glioma patients, as shown in Fig. 1B. Notably, the upregulation of NCF1, HSPB1, PIGT, PTX3, CCNA2, CCNB2, CCN4, and DES was significantly associated with poor prognostic outcomes. The gene expression heatmap, integrating clinical parameters, shows that higher expression levels of these genes correlate with advanced WHO grades, IDH wild-type status, 1p/19q non-codel status, and poor prognosis (Fig. 1C). Furthermore, BMP2, NCF1, HSPB1, PIGT, PTX3, CCNA2, and CCNB2 exhibited increased expression, while DES displayed decreased expression in glioma tissues (Fig. 1D). These gene expression patterns not only reflect the biological characteristics of the tumors but may also be pivotal for evaluating clinical outcomes. Thus, future research should focus on elucidating the functional roles of these genes and their mechanisms in tumor progression.

Figure 1 Construction of a prognostic model based on ISRGs.

(A) LASSO regression analysis was conducted on the genes associated with the IL18 signaling pathway. (B) The forest plot illustrates the prognostic significance of genes associated with the IL18 signaling pathway. (C) The heatmap illustrates the correlation between PTX3 and a range of clinical parameters. (D) The comparison of expression levels of immune-related signature genes (ISRGs) between the normal group and the glioma group. ***p < 0.001.

Expression correlation and prognosis of ISRGs

Subsequent investigations examined the interactions among these seven key genes in glioma. Using the Spearman correlation method, we identified significant correlations between these genes (Figs. 2A–2C). Additionally, Kaplan–Meier survival analyses confirmed the strong association between increased expression levels of NCF1, HSPB1, PIGT, PTX3, CCNA2, CCNB2, CCN4, and DES and poor prognostic outcomes in glioma patients. Notably, elevated expression of these genes is associated with shorter survival and disease progression, whereas increased BMP2 expression exhibited a contrasting effect (Fig. 2D).

Figure 2 The correlation and prognostic analysis of ISRGs.

(A) The chord diagram illustrates the relationships among significant genes. (B) The heatmap illustrates the relationships among significant genes. (C) The grid chart illustrates the relationship among significant genes. (D) The KM survival curves demonstrate notable variations in OS linked to nine essential genes. *p < 0.05.

PTX3 expression levels are significantly elevated in multiple cancers including glioma

The analysis of PTX3 expression across various cancer datasets provides strong evidence for its role in tumor biology. In our study, we observed an upregulation of PTX3 in gliomas, while its downregulation in 20 different cancer types suggests a potential tumor-suppressive role in these contexts (Figs. S1A, S1B). The correlation between PTX3 and immune infiltrating cells is noteworthy (Fig. S1C). PTX3 interacts with immune components, influencing tumor progression and the inflammatory microenvironment in cancers.

Functional enrichment analysis of PTX3 in glioma

The PPI network of DEGs was constructed using the STRING database to analyze their interactions, which were visualized with Cytoscape software (Fig. 3A). The interaction frequency of each gene is also shown (Fig. 3B). The results of the GO functional analysis and KEGG enrichment analysis are summarized below. The GO enrichment analysis identified significantly enriched biological processes, cellular components, and molecular functions, as detailed in Fig. 4 and Table 1. The key biological processes include leukocyte migration, leukocyte chemotaxis, myeloid leukocyte migration, and cytokine-mediated signaling pathways. The most prominent molecular functions include receptor–ligand activity, signaling receptor activator activity, cytokine activity, and extracellular matrix structural constituents. The cellular components with the highest enrichment levels are the secretory granule lumen, cytoplasmic vesicle lumen, vesicle lumen, and protein-DNA complex. Among the KEGG pathways, notable entries include cytokine-cytokine receptor interaction, transcriptional misregulation in cancer, the PI3K-Akt signaling pathway, and protein digestion and absorption (Figs. 4A, 4B) (Table 1). Additionally, GSEA revealed that the DEGs associated with PTX3 were significantly enriched in the integrin1, syndecan1, aurora-b, and IL18 signaling pathways (Figs. 5A–5C).

Figure 3 Construction of PPI network and identification of hub genes.

(A) The PPI network among DEGs. (B) The interaction number of each DEG.

Figure 4 Functional enrichment analysis of the DEGs based on the PTX3 expression levels.

(A) The chord diagram illustrates the results of the GO and KEGG enrichment analyses. (B) The circular illustration representing the enrichment analysis of GO and KEGG.

Table 1 Supplementary information of GO and KEGG analysis.

Ontology	ID	Description	adj. P. val	
BP	GO:0050900	Leukocyte migration	1.55616E−12	
BP	GO:0030595	Leukocyte chemotaxis	6.20629E−16	
BP	GO:0097529	Myeloid leukocyte migration	2.86071E−15	
BP	GO:0019221	Cytokine-mediated signaling pathway	2.3561E−05	
CC	GO:0034774	Secretory granule lumen	1.79515E−07	
CC	GO:0060205	Cytoplasmic vesicle lumen	2.07191E−07	
CC	GO:0031983	Vesicle lumen	2.27756E−07	
CC	GO:0032993	Protein-DNA complex	0.001199261	
MF	GO:0048018	Receptor ligand activity	5.80727E−15	
MF	GO:0030546	Signaling receptor activator activity	8.90369E−15	
MF	GO:0005125	Cytokine activity	5.66002E−13	
MF	GO:0005201	Extracellular matrix structural constituent	7.6857E−14	
KEGG	hsa04060	Cytokine-cytokine receptor interaction	3.66314E−10	
KEGG	hsa05202	Transcriptional misregulation in cancer	1.74594E−07	
KEGG	hsa04151	PI3K-Akt signaling pathway	0.000844153	
KEGG	hsa04974	Protein digestion and absorption	3.58154E−07	

Figure 5 GSEA of the altered signaling pathways in the glioma tissues.

(A) The bar chart illustrates the outcomes of GSEA. (B) The mountain plot illustrates the findings derived from the GSEA. (C) The analysis focuses on the enrichment of pathways associated with PTX3.

Association between the expression of PTX3 and survival outcomes in glioma

The mRNA expression of PTX3 in the validation datasets (GSE50161, GSE7696) was comparable. Upregulated mRNA levels of PTX3 were observed in tumor tissues across these datasets (Figs. 6A, 6D). The volcano plot clearly illustrates the differentially expressed genes. These findings suggest that PTX3 is differentially expressed in glioma, with elevated expression levels (Figs. 6B, 6E). Receiver operating characteristic (ROC) curve analysis further confirmed the robust performance of our risk model (Figs. 6C, 6F).

Figure 6 Verification and ROC curve of the PTX3 by GEO and CGGA datasets.

(A, D) The mRNA expression of PTX3 was analyzed using the GSE50161 and GSE7696 datasets. (B, E) The volcano plot emphasizes genes that demonstrate noteworthy upregulation and downregulation. (C, F) Diagnostic value of PTX3 in glioma as assessed by ROC curves. (G) Proportion of mortality as risk score values escalated within low and high-risk groups in CGGA. H High PTX3 expression was associated with poor OS in glioma in CGGA. (I–K) Higher levels of PT X3 expression are associated with more advanced WHO grade, IDH wild type, and 1p/19q non-codeletion in CGGA. **P < 0.01; ***P < 0.001.

To assess the reliability of the LASSO-derived prognostic model developed from the previous TCGA dataset, we utilized the CGGA dataset for external validation purposes. In addition, we analyzed the expression profiles of PTX3 in conjunction with risk scores, survival durations, and the distribution of survival statuses within the CGGA dataset. The analytical results demonstrated that PTX3 expression levels in the high-risk group were significantly higher than those in the low-risk group, correlating with poorer prognoses (Fig. 6G). The Kaplan–Meier (KM) curve analysis revealed a significant association between elevated expression levels of PTX3 and negative clinical outcomes (Fig. 6H). Additionally, the results showed that elevated PTX3 expression was associated with reduced survival rates in patients with G4, IDH wild-type, 1p/19q non-codel in patients diagnosed with glioma (Figs. 6I–6K).

Furthermore, we analyzed OS outcomes in patients categorized by low and high PTX3 expression levels across different phenotypes (Fig. 7). The results showed that elevated PTX3 expression was associated with reduced survival rates in patients with G3, G4, IDH wild-type, 1p/19q alterations, as well as in both female and male patients diagnosed with astrocytoma or glioblastoma. In contrast, no significant correlation was observed in patients classified as G2, IDH-mutated, oligoastrocytoma, or oligodendroglioma.

Figure 7 Kaplan–Meier survival analyses of glioma and its subtypes with different PTX3 expression levels.

PTX3 expression levels correlate with multiple clinicopathological characteristics in glioma

Significant differences were observed in WHO grade, IDH status, 1p/19q codeletion, age, histological classification, OS, DSS, and PFI between glioma patients with high and low PTX3 expression levels (Figs. 8A–8H). ROC curve analysis further validated the robust performance of our risk model (Figs. 8I–8L). The correlations between clinicopathological characteristics and PTX3 protein levels in glioma patients are presented in Table 2. Patients with high PTX3 expression exhibited poorer survival outcomes. We subsequently combined the WHO grade, IDH status, 1p/19q co-deletion, and PTX3 expression levels to develop a nomogram aimed at predicting survival outcomes (Fig. 9A). Importantly, the expression of PTX3 has shown promise in improving the precision of survival probability estimations at the 1-, 3-, and 5-year marks. Furthermore, a calibration chart was employed to assess the accuracy of the predictions generated by the model (Fig. 9B). The time-dependent ROC curves revealed AUC values for the 1, 3, and 5-year periods that surpassed 0.75, signifying the robust performance of the model (Fig. 9C). Univariate Cox regression analysis identified WHO grade, IDH status, 1p/19q codeletion, age, and PTX3 expression as independent prognostic markers for glioma patients. Furthermore, multivariate Cox regression analysis confirmed WHO grade, IDH status, and age as independent prognostic determinants (Fig. 9D).

Figure 8 Association between PTX3 expression and clinical features.

(A–H) The correlation between PTX3 expression and multiple clinical parameters—such as the WHO grading system, IDH mutation status, 1p/19q codeletion status, patient age, histological classification, OS events, DSS events, and PFI events—has been explored. (I–L) The parameters were analyzed by ROC curves. ***P < 0.001.

Table 2 Clinicopathological characteristics of glioma patients with high- and low-PTX3 expression levels.

Characteristics	Low expression of PTX3	High expression of PTX3	P value	
n	349	350		
WHO grade, n (%)			<0.001	
G2	171 (26.8%)	53 (8.3%)		
G3	125 (19.6%)	120 (18.8%)		
G4	8 (1.3%)	160 (25.1%)		
IDH status, n (%)			<0.001	
WT	27 (3.9%)	219 (31.8%)		
Mut	320 (46.4%)	123 (17.9%)		
1p/19q codeletion, n (%)			<0.001	
Non-codel	223 (32.2%)	297 (42.9%)		
Codel	126 (18.2%)	46 (6.6%)		
Age, n (%)			<0.001	
<= 60	311 (44.5%)	245 (35.1%)		
>60	38 (5.4%)	105 (15%)		
OS event, n (%)			<0.001	
Alive	280 (40.1%)	147 (21%)		
Dead	69 (9.9%)	203 (29%)		
DSS event, n (%)			<0.001	
No	284 (41.9%)	150 (22.1%)		
Yes	61 (9%)	183 (27%)		
PFI event, n (%)			<0.001	
No	228 (32.6%)	125 (17.9%)		
Yes	121 (17.3%)	225 (32.2%)		

Figure 9 PTX3 expression prognostic analysis.

(A) The nomogram is constructed by combining pathological features and PTX3 expression. (B) The calibration chart illustrates the predictive accuracy determined through multi-factor Cox regression analysis. (C) Time-dependent ROC curves indicated the good performance of our prognostic model. (D) Validation of overall survival OS analysis in glioma using Cox regression from the TCGA database.

TF-gene interaction and TF-miRNA coregulatory network

The NetworkAnalyst database facilitated the prediction and visualization of interactions between transcription factors (TFs) and hub genes. The resulting TF-gene interaction network comprised 18 nodes and 19 edges, as illustrated in Fig. 10A. Specifically, FOXL1 demonstrated interactions with two hub genes, namely PTX3 and CXCL8, whereas TP53 was found to interact with PTX3 and MMP9. Nonetheless, these observations necessitate additional validation. Following this, the NetworkAnalyst tool was employed to develop the TF-miRNA co-regulatory network, depicted in Fig. 10B.

Figure 10 TF-gene interaction and TF-miRNA coregulatory network.

(A) Network for TF-gene interaction with three hub genes. (B) The network presents the TF-miRNA coregulatory network.

PTX3 expression levels in tissue samples of glioma patients

To assess the differences in PTX3 protein expression between adjacent paracancerous and glioma tissues, we employed IHC to examine the expression levels of PTX3 in both tissue types. HE staining revealed that tumor cells in LGG displayed a small, rounded, adipose-like morphology, characterized by invasive growth. These cells exhibited a mild increase in density, substantial cytoplasm, and infrequent mitotic figures. In contrast, tumor cells in GBM were marked by invasive growth and a rounded shape, with a moderate to high increase in cell density and reduced cytoplasmic volume. Mitotic activity was notably high, with over 20 mitoses observed per 10 high-power fields (HPF), along with significant pseudopalisading necrosis and endothelial proliferation. IHC results showed that PTX3 expression was significantly higher in glioma tissues compared to adjacent paracancerous tissues (Fig. 11A). Furthermore, we assessed the relationship between PTX3 expression and various clinical parameters in patients. Elevated levels of PTX3 expression were found to be associated with a more advanced WHO grade, non-codel status of 1p/19q, as well as IDH wild type (Figs. 11B–11E).

Figure 11 Assessment of PTX3 expression in clinical specimens from glioma patients.

(A) The expression levels of the PTX3 protein were elevated in glioma tissues when contrasted with normal brain tissues. (B–D) The elevated level of PTX3 expression was correlated with more advanced WHO grade, IDH wild type, and 1p/19q. (E) The circular diagram depicts the correlation between the expression levels of PTX3 and a range of clinical pathological characteristics linked to glioma. *P < 0.05; **p < 0.01, ***p < 0.001.

Biological significances of PTX3 in glioma progression

In order to evaluate the function of PTX3 in glioma, a series of in vitro experiments were conducted. qPCR analysis demonstrated that the suppression of PTX3 resulted in a reduction of its mRNA expression levels in U251 cells (Fig. 12A). Furthermore, a significant increase in the number of clones was observed in the si-NC group compared to the si-PTX3 group (Figs. 12B, 12C). The results from the CCK-8 assay indicated that the proliferative capacity of U251 cells was markedly diminished following the knockdown of PTX3 (Fig. 12D). Additionally, wound healing assays indicated a decline in the metastatic potential of U251 cells subsequent to PTX3 knockdown (Figs. 12E, 12F). The inhibition of glioma cell growth and migration upon PTX3 knockdown suggests that PTX3 plays a role in promoting glioma progression.

Figure 12 Knockdown of PTX3 inhibited glioma progression.

(A) The knockdown of PTX3 resulted in decreased mRNA levels of PTX 3 in the U251 cell, as confirmed by qPCR analysis. (B, C) The knockdown of PTX3 suppresses the proliferation of U251 cell, as evidenced by clone formation assays. (D) The knockdown of PTX3 suppresses the proliferation of U251 cell, as evidenced by CCK-8 assays. (E, F) The knockdown of PTX3 inhibits the migration of U251 cell, as demonstrated by wound healing assays. **p < 0.01, ***p < 0.001.

Discussion

Glioma is an aggressive brain tumor that poses significant challenges in treatment and prognosis, primarily due to its resistance to conventional therapies and its high mortality rate (Andrew Awuah et al., 2024; Skouras et al., 2024). Identifying reliable prognostic biomarkers is crucial for improving patient outcomes and optimizing treatment strategies. Although various biomarkers have been identified in current research, the specific role of ISRGs in glioma remains underexplored. This study aims to address this gap by systematically analyzing the expression profiles of ISRGs and their correlation with clinical features, thereby providing new insights into the prognostic patterns of glioma and emphasizing the need for further investigation into the functional significance of these genes in tumor biology.

IL-18, a pro-inflammatory cytokine primarily secreted by macrophages and dendritic cells, plays a crucial role in regulating immune responses and inflammation (Cheng et al., 2024; Kessel, Rossig & Abken, 2025). In the glioma microenvironment, overexpression of IL-18 often leads to enhanced local inflammation, which promotes tumor cell proliferation and invasive behavior (Ji et al., 2024; Yeh et al., 2012). Therefore, changes in IL-18 levels can serve as an indicator of disease severity and prognosis in glioma patients (Sun et al., 2023). Furthermore, the IL-18 signaling pathway plays a significant role in glioma development by influencing tumor immune evasion mechanisms (Sun et al., 2023). Studies have shown that IL-18 enhances the activity of cytotoxic T lymphocytes and natural killer (NK) cells, thereby boosting the immune recognition of tumors (Street et al., 2004). IL-18 serves not only as a potential prognostic marker for glioma progression but also as a promising therapeutic target due to its role in immune modulation. By elucidating the function of IL-18 and its associated signaling pathways, we aim to contribute to the development of targeted interventions that may improve treatment outcomes for glioma patients.

This study aimed to identify prognostic biomarkers associated with glioma, focusing on the expression of ISRGs and their correlation with clinical features. By employing a comprehensive approach, including RNA-seq data analysis, LASSO Cox regression, and immunohistochemistry, we developed a gene signature consisting of nine key genes: BMP2, NCF1, HSPB1, PIGT, PTX3, CCNA2, CCNB2, CCN4, and DES. Our findings revealed a significant association between the expression levels of these genes and overall survival in glioma patients, emphasizing the potential of ISRGs as valuable prognostic indicators. These results highlight the need for further investigation into the functional roles of these genes and their impact on glioma progression, laying the foundation for future research aimed at improving patient prognosis. Notably, the upregulation of several of these genes, particularly NCF1, HSPB1, PIGT, and PTX3, was associated with poor prognosis, supporting the idea that these genes play a pivotal role in tumor progression and patient survival. Our findings further emphasize the critical role of PTX3 in glioma, particularly regarding patient prognosis and immune infiltration. PTX3, a member of the pentraxin family, is well-known for its involvement in immune responses and has been linked to various cancers. The upregulation of PTX3 in glioma tissue correlates with poor clinical outcomes, suggesting its role in promoting the invasiveness of this malignancy. This finding is significant as it underscores the potential of PTX3 as a prognostic marker, which could assist in stratifying patients based on risk and adjusting treatment strategies accordingly.

Functional enrichment analysis revealed that DEGs associated with PTX3 are significantly enriched in several key pathways, including cytokine-cytokine receptor interactions, the PI3K-Akt signaling pathway, and transcriptional dysregulation in cancer. The cytokine-cytokine receptor interaction pathway plays a critical role in mediating immune responses and inflammation, which are essential in the tumor microenvironment (Bhat et al., 2021; Pei, Gao & Wu, 2024). This pathway facilitates communication between tumor cells and immune cells, potentially influencing tumor progression and treatment response. In this context, the upregulation of PTX3 may enhance immune cell recruitment and activation, thereby modulating the immune environment in glioma and providing a rationale for immune-modulatory therapeutic strategies. The PI3K-Akt signaling pathway, another key pathway identified in our analysis, is well known for its role in regulating cell proliferation, survival, and metabolism (Asati, Mahapatra & Bharti, 2016; Hoxhaj & Manning, 2020). Dysregulation of this pathway is frequently observed in various cancers, including glioma, and is associated with poor prognosis. The association between PTX3 and this pathway suggests that PTX3 may contribute to glioma cell survival and proliferation, highlighting its potential as a therapeutic target. Combined with PTX3 regulation, targeting the PI3K-Akt pathway may offer a novel approach to enhance therapeutic efficacy in glioma patients. Overall, these pathways not only illuminate the molecular mechanisms through which PTX3 may influence glioma biology but also underscore its potential as a therapeutic target for the development of more effective treatment strategies.

Our immune analysis revealed a significant correlation between PTX3 expression and the infiltration of various immune cells in glioma tissues. Macrophages are well-known for their plasticity and can adopt either pro-inflammatory or anti-inflammatory phenotypes depending on the tumor microenvironment. The positive correlation between PTX3 and macrophage infiltration suggests that PTX3 may promote the recruitment or activation of these immune cells, thereby enhancing anti-tumor immunity. Previous studies have shown that tumor-associated macrophages (TAMs) can promote tumor growth and metastasis, but when appropriately activated, they can also mediate tumor-killing effects (Parveen et al., 2021; Sun et al., 2021). Therefore, understanding the specific role of PTX3 in macrophage polarization and function may provide insights into therapeutic strategies aimed at reprogramming macrophages to adopt more anti-tumor phenotypes. The positive correlation between PTX3 and neutrophils is also noteworthy. While neutrophils can exert anti-tumor effects, they can also promote tumor progression by releasing pro-inflammatory cytokines and forming neutrophil extracellular traps (NETs) (Di Mambro et al., 2023; Opoku et al., 2021). The correlation between PTX3 and neutrophil infiltration suggests that PTX3 may influence neutrophil behavior in the glioma microenvironment, potentially affecting tumor dynamics and patient prognosis. In summary, the immune cell infiltration patterns associated with PTX3 expression highlight its potential as a key regulator of the glioma immune microenvironment. Further investigation is essential to elucidate the mechanisms by which PTX3 influences immune cell dynamics and to explore its feasibility as a therapeutic target in glioma treatment strategies.

The diagnostic paradigm for the molecular classification of gliomas incorporates the WHO grade, IDH status, and the presence of 1p/19q co-deletion. Our findings demonstrate a significant association between PTX3 expression levels and survival outcomes in glioma patients, highlighting its prognostic value across various clinical phenotypes. Increased PTX3 expression correlates with higher WHO grades (G3 and G4), IDH-wildtype status, and 1p/19q alterations, all of which are associated with decreased overall survival. The differential expression of PTX3 across glioma subtypes suggests its potential role in regulating tumor behavior and patient outcomes through underlying biological mechanisms. Furthermore, univariate and multivariate Cox regression analyses identified PTX3 as an independent prognostic marker, alongside other established factors such as WHO grade, IDH status, and age. This reinforces the potential value of PTX3 as a supplementary factor in clinical decision-making, particularly in risk stratification and treatment tailoring based on patient characteristics. The absence of significant correlation in lower-grade gliomas (G2) and IDH-mutant cases suggests that the prognostic significance of PTX3 is more pronounced in aggressive tumor phenotypes, warranting further investigation into its mechanistic role in glioma pathophysiology.

However, this correlation implies that PTX3 expression may be influenced by these categorical determinants, potentially compromising its prognostic significance in multivariate analyses. Within this framework, these molecular markers may serve both as confounding and mediating variables. Additionally, elderly patients are at an increased risk of blood–brain barrier impairment, which heightens the likelihood of PTX3 derived from peripheral blood infiltrating the tumor microenvironment (Zhang et al., 2022). Furthermore, the phenomenon of immune aging may also facilitate tumor advancement. This interplay establishes a “common cause” relationship between exposure and outcome, rendering age a notable confounding variable. However, conventional Cox proportional hazards models are limited to adjusting for linear effects, inadequately capturing the intricate, time-dependent nature of interactions with the microenvironment. Treatment responses can alter PTX3 expression, such as through the activation of the NF-κB signaling pathway via mechanisms related to DNA damage repair and subsequent survival outcomes. Nevertheless, omitting treatment data from the model may yield residual confounding biases. Variations in PTX3 expression within glioblastomas may occur between the tumor core and the infiltrative border. Moreover, the specific sites of biopsy sampling can introduce classification errors. Transcriptomic analyses are unable to differentiate between PTX3 produced by tumor cells and that secreted by peritumoral microglia. This complexity in cell-type specific expression necessitates investigation through spatial multi-omics methodologies. Consequently, future research must address these challenges. In conclusion, our survival analysis not only elucidates the prognostic implications of PTX3 and its associated gene network but also paves the way for further investigation into their roles within glioma biology. Subsequent studies should prioritize the validation of these findings in larger cohorts and the exploration of the underlying mechanisms through which PTX3 influences glioma progression and patient outcomes. Such endeavors may ultimately enhance our comprehension of glioma pathophysiology and yield essential insights for the development of targeted therapeutic strategies.

In summary, this study highlights the critical role of PTX3 in glioma, underscoring its potential as both a prognostic biomarker and a therapeutic target. The integration of bioinformatics methods with clinical data facilitated the identification of key IL-18 signaling-related genes associated with poor patient outcomes, particularly PTX3. The correlation between PTX3 expression and immune cell infiltration further emphasizes its involvement in the tumor microenvironment and immune regulation. Future research should focus on the functional role of PTX3 and its interactions within the glioma microenvironment, which could lead to novel therapeutic strategies for improving patient management based on its expression profile.

Conclusions

This study effectively combines macroscopic and microscopic approaches to trace the screening trajectory of ISRGs. Utilizing techniques such as extensive data analysis, bioinformatics, and histopathological examination, ensures a comprehensive investigation of ISRGs. The findings demonstrate that elevated PTX3 expression in glioma is associated with disease progression, poor prognostic outcomes, and abnormal immune cell infiltration. We anticipate that identifying PTX3 will enhance personalized treatment strategies for glioma patients and support more informed treatment decisions.

Supplemental Information

Supplemental Information 1 Raw data

Supplemental Information 2 256 IL18 signaling-related genes

Supplemental Information 3 The expression pattern of PTX3 in different samples

(A) The comparative analysis of PTX3 expression levels in normal tissues versus various pan-cancer samples. (B) The radar chart illustrates the relative expression levels of PTX3 in different human cancerous tissues when juxtaposed with normal tissue counterparts. (C) The relationship between PTX3 expression and immune cell infiltration across 33 distinct cancer tissue types. *P < 0.05; **P < 0.01; ***P < 0.001.

Supplemental Information 4 R Code

Additional Information and Declarations

Competing Interests

Author Contributions

Human Ethics

Data Availability

The authors declare there are no competing interests.

Delin Wang conceived and designed the experiments, analyzed the data, prepared figures and/or tables, authored or reviewed drafts of the article, and approved the final draft.

Cuimei Liu conceived and designed the experiments, authored or reviewed drafts of the article, and approved the final draft.

Bohao Sun conceived and designed the experiments, performed the experiments, analyzed the data, prepared figures and/or tables, authored or reviewed drafts of the article, and approved the final draft.

Xiaodong Zhang performed the experiments, analyzed the data, prepared figures and/or tables, authored or reviewed drafts of the article, and approved the final draft.

Yejun Zhou performed the experiments, analyzed the data, prepared figures and/or tables, authored or reviewed drafts of the article, and approved the final draft.

Zhonglin Hu analyzed the data, prepared figures and/or tables, authored or reviewed drafts of the article, and approved the final draft.

Duanzheng Cao analyzed the data, authored or reviewed drafts of the article, and approved the final draft.

Jing Zhang conceived and designed the experiments, performed the experiments, prepared figures and/or tables, authored or reviewed drafts of the article, and approved the final draft.

Jinfang Xu conceived and designed the experiments, prepared figures and/or tables, authored or reviewed drafts of the article, and approved the final draft.

The following information was supplied relating to ethical approvals (i.e., approving body and any reference numbers):

Ethical approval was received for the study by the Clinical Research Ethics Committee of The Second Affiliated Hospital, Zhejiang University School of Medicine (approval no. 2024-0649).

The following information was supplied regarding data availability:

The raw measurements are available in the Supplementary File.

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
