# Peer review of "Expression of the IL-18-related gene PTX3 correlates with clinicopathological features and prognosis in glioma patients"

_PeerJ, doi:10.7717/peerj.19675_

## Round 0.1 · original submission · Major Revisions

Reviewer 1 ·

Basic reporting

Clarity and Language: While the manuscript is generally well-written, some sections (e.g., methods and results) are overly verbose, making key findings difficult to discern. For example, the description of statistical analyses lacks conciseness.

Figures and Tables: Several figures (e.g., Figures 1–13) are referenced but not provided in the submission, hindering a thorough evaluation of data quality and interpretation.

Experimental design

Lack of Functional Validation: The study relies heavily on bioinformatics analyses and immunohistochemistry but fails to include in vitro or in vivo experiments to validate the mechanistic role of PTX3 in glioma progression or immune modulation. Correlative findings without functional evidence weaken the conclusions.

Sample Size Limitations: The immunohistochemistry validation cohort (n=56) is underpowered, particularly for subgroup analyses (e.g., WHO grade, IDH status). This raises concerns about the generalizability of the results.

Model Validation: The LASSO-derived prognostic model lacks validation in an independent external cohort, casting doubt on its reproducibility and clinical applicability.

Validity of the findings

Overinterpretation of Correlations: The association between PTX3 expression and immune cell infiltration is presented as causal, yet no experimental evidence (e.g., cytokine assays, co-culture experiments) supports this claim. The observed correlations may reflect bystander effects rather than direct biological roles.
Pan-Cancer Analysis Distraction: The inclusion of pan-cancer PTX3 expression data dilutes the focus on glioma and lacks clear relevance to the study’s primary aims.

Additional comments

This study identifies intriguing correlations between PTX3 and glioma prognosis but falls short of providing mechanistic insights or robust validation. The findings, while potentially significant, remain preliminary. Rejection is recommended in its current form. The authors should address the above concerns through functional experiments, independent cohort validation, and a more focused analysis of PTX3’s role in glioma before resubmission.

Reviewer 2 ·

Basic reporting

The manuscript investigates the role of PTX3 in glioma, focusing on its correlation with clinical features, immune infiltration, and prognosis. The study combines bioinformatics analysis (TCGA/GTEx data) with experimental validation (IHC on 56 glioma specimens). The findings suggest PTX3 as a potential prognostic biomarker and therapeutic target. While the study is well-structured and addresses a relevant gap in glioma research, several issues need clarification or improvement before acceptance.

Strengths: a. Clear hypothesis and rationale for studying PTX3 in glioma. b. Comprehensive methods (LASSO regression, functional enrichment, and immune infiltration analysis). c. Raw data and validation cohorts (GSE50161, GSE7696) are provided.

Concerns: a. Language/Clarity: Some sentences are overly complex (e.g., Introduction, lines 64–73). Simplify for readability. b. Figures/Tables: Figure 1D: Clarify why DES shows decreased expression in glioma despite its association with poor prognosis. Table 2: Include statistical tests used for comparisons (e.g., chi-square for categorical data).

The manuscript presents valuable insights, but requires:
1. Clarification of mechanistic limitations (lack of functional experiments).
2. Justification for the small IHC cohort.
3. Improved language clarity and statistical reporting.
4. Discussion of confounding factors in survival analysis.

Experimental design

Strengths: a. Robust use of public datasets (TCGA, GTEx) and validation cohorts. b. Multifaceted approach (bioinformatics + IHC).

Concerns: a. Sample Size: The IHC cohort (n=56) is small. Justify the sample size or acknowledge limitations. b. Ethics: Ethical approval is noted, but confirm if patient consent was waived for retrospective data (line 194).

Validity of the findings

Strengths: a. PTX3’s association with immune infiltration (macrophages, T cells) is compelling and aligns with prior literature. b. The prognostic model (9-gene signature) is statistically validated.

Concerns: a. Mechanistic Insight: The study correlates PTX3 with poor prognosis but does not experimentally validate its functional role (e.g., knockdown/overexpression in glioma cells). Suggest future work. b. Confounding Factors: Did the authors adjust for covariates (e.g., treatment history) in survival analyses?

Additional comments

a. References: Some citations are outdated (e.g., Choi et al. 2011). Include recent studies (2023–2025) on PTX3 in cancer.

b. Abbreviations: Define all abbreviations at first use (e.g., ssGSEA, DSS).

---

## Round 0.2 · accepted · Accept

I confirmed that the authors have addressed all of the reviewers' comments. The manuscript is ready for publication.

Reviewer 2 ·

Basic reporting

no comment

Experimental design

no comment

Validity of the findings

no comment

Additional comments

no comment